# Domain Shared and Specific Prompt Learning for Incremental Monocular Depth Estimation

## ABSTRACT

Incremental monocular depth estimation aims to continuously learn from new domains while maintaining their performance on old domains. The catastrophic forgetting problem is the key challenge when the model adapts the dynamic scene variations. Previous methods usually address this forgetting problem by storing raw samples from the old domain, allowing the model to review the knowledge of the old domain. However, due to the concerns of data privacy and security, our objective is to tackle the incremental monocular depth estimation problem in more stringent scenarios without the need for replaying samples. In this paper, we attribute the cross-domain catastrophic forgetting to the domain distribution shifts and continuous variations of depth space. To this end, we propose **D**omain **S**hared and **S**pecific **P**rompt Learning (DSSP) for incremental monocular depth estimation. In detail, to alleviate the domain distribution shift, complementary domain prompt is designed to learn the domain-shared and domain-specific knowledge which are optimized by the inter-domain alignment and intra-domain orthogonal loss. To mitigate the depth space variations, we first introduce a pre-trained model to generate the domain-shared depth space. Then, we design $S^2$-Adapter that quantizes depth space variations with scale&shift matrices and converts the domain-shared depth space to domain-specific depth space. Our method achieves state-of-the-art performance under various scenarios such as different depth ranges, virtual and real, different weather conditions, and the few-shot incremental learning setting on 12 datasets. We will release the source codes and pre-trained models.

## CCS CONCEPTS

• **Computing methodologies** → **Computer vision**; **Lifelong machine learning**; **Scene understanding**.

## KEYWORDS

Monocular depth estimation, incremental learning, domain prompt learning

## 1 INTRODUCTION

Monocular depth estimation aims to generate dense depth maps from a single RGB image, which is an important scene perception technique. In practice, new domains substantially emerge with

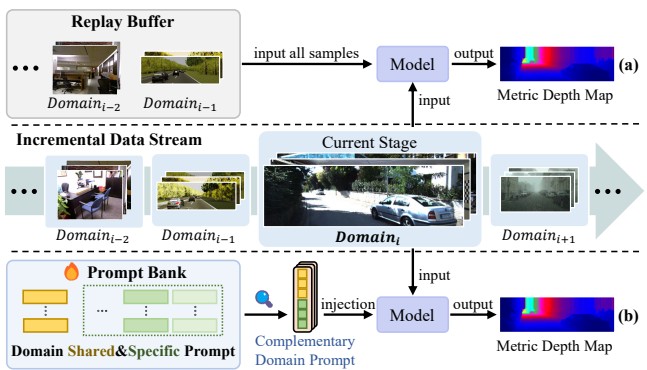

**Figure 1: Overview of our method (b). Compared with typical approaches (a) that rely on rehearsal buffers to alleviate forgetting, DSSP mitigates domain distribution shifts and depth space variations through the complementary domain prompt and $S^2$-Adapter.**

environmental variations, such as weather and illumination conditions. Incremental monocular depth estimation intends to facilitate continuous learning and adaptation of the model in such dynamic environments while maintaining its depth estimation performance in the old domain.

In incremental monocular depth estimation, the catastrophic forgetting problem is the key challenge that is caused by domain distribution shifts and depth space variations. 1) Each domain has a unique data distribution corresponding to the environmental factors, which leads to severe domain gaps among them. Consequently, when the model adapts to a new domain, the shift of feature distribution leads to the phenomenon of forgetting old knowledge. 2) Different domains usually have different depth ranges and spatial structures, so the depth space exhibits scale and shift variations across domains. When the model adapts to these variations on new domains, it may result in the forgetting of old knowledge.

To address the aforementioned problems, some approaches [28, 29] attempt to reduce the domain gaps by employing multi-domain learning across as many domains as possible and alleviate the influence of the depth space variations by predicting relative depth maps. However, exhaustively covering all domains in the real world is impractical, and retraining the model from scratch consumes plenty of time and computational resources. Other methods [6, 11, 40] try to continuously fine-tune the model and ease the forgetting problem by maintaining a replay buffer involving samples from old domains which enables the model to review them while learning new knowledge. Such as LL-MonoDepth [11] stores 500 samples in each domain for reviewing old knowledge while employing multiple predictors to adapt the depth space variations on different domains. However, storing samples may not always be feasible,

especially for scenarios where long-term storage of data is not permitted owing to privacy or data use legislation [34]. Therefore, we aim to address the incremental monocular depth estimation in a more strict protocol without the need for replay samples.

Recently, pre-trained models embed knowledge into enormous parameters, and this empowers them with a better capability of mitigating forgetting. Researchers leverage prompt learning with pre-trained models to learn task-specific knowledge in the new domain. Inspired by this, we introduce learnable prompts to capture and retain domain-specific knowledge, which represents domain distribution information through several parameters, thereby saving storage costs and avoiding concerns of data privacy and security. Furthermore, prompts can also learn unified feature representations across domains to reduce domain gaps and enhance the generalization capability of the model.

Therefore, we propose **D**omain **S**hared and **S**pecific **P**rompt Learning (DSSP) for incremental monocular depth estimation, which designs complementary domain prompts and Scale&Shift Adapters ($S^2$-Adapter) to mitigate domain distribution shifts and depth space variations. Specifically, we first design domain-shared prompts to continually learn across all domains and prevent knowledge forgetting by domain shift through inter-domain alignment constraint, while we design domain-specific prompts solely to learn in the corresponding domain to enhance the adaptability of the model and retain knowledge in their parameters. Additionally, we introduce an intra-domain orthogonal loss to constrain these two types of prompts to focus on the unified inter-domain representation and individual domain distribution respectively. Both two types of prompts are maintained in the prompt bank. Therefore, we leverage the frozen pre-trained model with the complementary domain prompts to predict a relative depth map. Furthermore, to capture the depth space variations and convert the relative depth into metric depth, we devise the lightweight $S^2$-Adapter which precisely learns the map between the relative depth and metric depth through quantifying depth space variations into scale&shift matrices. As illustrated in Figure 1, in contrast to previous replay-based methods, DSSP generates complementary domain prompts by learning domain-shared prompts and domain-specific prompts, facilitating the learning of new knowledge while retaining previous knowledge without data privacy and security concerns.

During the inference phase, we predict the domain of the test image according to the domain-specific prompt. Then, we generate complementary domain prompt and subsequently inject them into the pre-trained model to predict the relative depth map. Next, the $S^2$-Adapter is selected by the domain identity to restore the depth space transformation and predict the final depth. Experiments on 12 datasets demonstrate our DSSP outperforms the SOTA methods.

The main contribution of our work can be summarized as follows:

- We propose domain shared and specific prompt learning to alleviate the catastrophic forgetting problem of incremental monocular depth estimation without data privacy and security concerns.
- We devise a $S^2$-Adapter which quantifies depth space variations into scale&shift matrices to capture the map between relative depth and metric depth and mitigate the influences of depth space variations.

- Extensive experiments on multiple domain incremental sequences show our method achieves superior incremental monocular depth estimation performance.

## 2 RELATED WORK

### 2.1 Monocular Depth Estimation

As a technique for scene perception, monocular depth estimation aims to generate per-pixel depth maps for scenes based on a single RGB image. Previous learnable methods achieve satisfactory performance through supervised [1, 9, 12, 14, 15, 19, 21, 27], self-supervised [10, 20, 24, 39, 42, 51] and unsupervised learning [36, 49, 50, 52] on single domain. However, due to the lack of robustness and generalization, models usually exhibit poor performance in new domains when they are applied in the real world. Some works tend to collect a large scale of data samples across various domains and learn a domain-invariant model [29, 37, 43, 46, 47]. However, it is impossible to cover all domains in practical applications. Therefore, when a new domain emerges, the model has to combine the data of the new domain with all old domains and be retrained from scratch, which is time-consuming. Meanwhile, the large number of parameters in these pre-trained models makes training them from scratch expensive in computing resources. To address the above problem, researchers devote numerous efforts to exploring incremental monocular depth estimation methods, so that the model can continuously adapt to the new domain without forgetting the knowledge of the old domain. Recently, some researchers try to store a part of samples of old domains and maintain a replay buffer that allows the model to access when fine-tuning on the new domain. Although the model alleviates the catastrophic forgetting problem by reviewing these replay samples, it is not practical to store data for a long time due to data privacy and security concerns in many real-world scenarios.

### 2.2 Incremental Learning

Incremental Learning is a typical setting to continuously train a single model on non-stationary data distributions. The major challenge is the catastrophic forgetting problem [26], where adaptation to a new distribution usually results in a much-reduced ability to capture the old distribution. Inspired by the complementary learning systems (CLS) theory [18, 25], researchers devote numerous efforts to facilitating the model to learn from continuous data streams, such as storing some training samples for each of the previous tasks and reusing them while learning a new task (replay methods) [2, 3, 5, 23, 30, 41, 48]; imposing extra regularization instead of storing training samples (regularization-based methods), such as LwF [22] using a knowledge distillation loss on previous tasks and EWC [17] enforcing an additional loss term to alleviate changing on the weights important for previous tasks; fixing trained parameters on old tasks and employ extra network branches for training a new task (parameter isolation methods) [16, 31, 33, 44]. These methods are empirically demonstrated useful for image recognition [16, 17, 41], while they mainly focus on the class-incremental problem. Directly transferring these methods to incremental monocular depth estimation may face two challenges, domain distribution shifts and depth space variations.

Figure 2: Overview of the proposed network architecture. We propose domain shared and specific prompt learning for alleviating domain distribution shifts and $S^2$-Adapter for adapting the depth space variations.

## 3 MYTHOLOGY

In this section, we first introduce the paradigm of incremental monocular depth estimation tasks. Then, we overview our pipeline during both the training and inference phases. Next, we describe the complementary domain prompt learning strategy. Furthermore, we detail the $S^2$-Adapter. Finally, we elaborate on inferring the domain identity of the test images for the generation of complementary domain prompt and the selection of $S^2$-Adapter.

### 3.1 Incremental Monocular Depth Estimation

In this task, the data of different domains are incrementally obtained. We define a sequence $\mathcal{D}$ with $S$ domains, $\mathcal{D} = \{\mathcal{D}_1, \cdots, \mathcal{D}_S\}$. In the $s$-th domain $\mathcal{D}_s = \{(x_i^s, y_i^s)\}_{i=1}^{N_s}$, there are $N_s$ images $x_i^s \in \mathcal{X}^s$ and its corresponding depth map $y_i^s \in \mathcal{Y}^s$. Given an image $x$ from arbitrary domains, our goal is to train a single model $f_{\theta_s} : \mathcal{X}^s \to \mathcal{Y}^s$ parameterized by $\theta_s$ to predict the depth map $\hat{y}$ denotes as $\hat{y} = f_{\theta}(x)$. Generally, the model $f_{\theta_s}$ is optimized by,

$$\mathcal{L} = \mathcal{L}_d(y, \hat{y}) + \mathcal{R} \tag{1}$$

where $\mathcal{L}_d$ is the depth estimation loss to learn the knowledge on the new domain, and $\mathcal{R}$ is a regularization loss for preserving the knowledge from old domains. Our method tackles the more challenging incremental monocular depth estimation which data from previous domains are not be seen anymore.

### 3.2 Overview

In incremental monocular estimation, the catastrophic forgetting problem is caused by continuous domain distribution shifts and depth space variations. To this end, we propose DSSP, which is composed of complementary domain prompt and $S^2$-Adapter. As illustrated in Figure 2, we maintain a prompt bank containing learnable prompts throughout all incremental stages. The complementary domain prompt, along with image embedding, is fed into the frozen pre-trained depth estimation model and then yields a relative depth map. We froze the pre-trained model to prevent it from the influenced by domain distribution variations. Subsequently, following adjustments in scale and shift by our $S^2$-Adapter, the relative depth map is transformed into the final metric depth map. In the inference phase, we first infer the domain identity of the input image, thereby generating the complementary domain prompt and selecting the $S^2$-Adapter based on the domain identity. Finally, based on the domain distribution knowledge within the prompt and the depth space rule within the $S^2$-Adapter, we can continuously estimate the depth map accurately with less forgetting. During all incremental learning stages, only the complementary domain prompt and the $S^2$-Adapter are trainable, which only cost less than 1M parameter per domain.

### 3.3 Complementary Domain Prompt Learning

To address the issue of forgetting in incremental monocular depth estimation, previous methods store raw samples from the old domain and input them alongside samples from the new domain into the model. These approaches allow the model to review knowledge from the old domain while learning from the new domain. However, in many real-world scenarios, data from the old domain may not be retained and accessed in the long term due to privacy

and security concerns. Therefore, instead of directly storing raw samples of each domain, we introduce learnable prompts to capture and store domain knowledge. These learnable prompts can explore the individual features of each domain, enhancing the adaptability of pre-trained models. Meanwhile, these prompts can also mine inter-domain common knowledge through long-term learning to improve the cross-domain generalization of the model. Overall, the prompts can not only learn the characteristics of each domain but also model generic information across domains, thereby mitigating the catastrophic forgetting problem without data privacy and security concerns.

Based on the aforementioned analysis, we devise complementary domain prompt learning, which consists of domain-shared prompt and domain-specific prompt to learn both domain-general knowledge and domain-specific knowledge for each domain respectively. Given a pre-trained model with $M$ transformer layers in which the embedding dimension of the hidden layer is $d$, we learn $M$ domain-shared prompt $\mathbf{P}_{shared}$ and domain-specific prompt $\mathbf{P}_{specific}$:

$$\mathbf{P}_{shared} = \{p_{shared}^i \in \mathbb{R}^{l_p \times d} | 0 \le i \le M - 1\}; \quad (2)$$

$$\mathbf{P}_{specific} = \{p_{specific}^i \in \mathbb{R}^{l_p \times d} | 0 \le i \le M - 1\} \quad (3)$$

where $l_p$ is the length of prompts. Then, we generate the complementary domain prompt $\mathbf{P}_{comp} = \{p_i \in \mathbb{R}^{2*l_p \times d} | 0 \le i \le M - 1\}$ by:

$$\mathbf{P}_{comp} = [\mathbf{P}_{shared}, \mathbf{P}_{specific}], \quad (4)$$

where $[\cdot, \cdot]$ denotes the concatenation along the length dimension. Next, we inject the complementary domain prompt $\mathbf{P}_{comp}$ into the first transformer layer $L_1$ of the pre-trained model alongside the image embedding $e$ which is extracted by the input image $I$:

$$[cls_1, \_, h_1] = L_1([cls_0, p_0, e]), \quad (5)$$

Subsequently, the rest complementary domain prompt is injected into the pre-trained model with each hidden embedding feature:

$$[cls_i, \_, h_i] = L_i([cls_{i-1}, p_{i-1}, h_{i-1}]) \quad i = 2, 3, \dots, M. \quad (6)$$

where $cls_i$ is the class token, and $h_i \in \mathbb{R}^d (1 \le i \le M - 1)$ is the hidden embedding feature output from the layer $L_i$.

During training, the domain-shared prompt continuously learns domain generalized knowledge across all domains. In order to prevent the domain-shared prompt from forgetting information about old domains, we introduce an inter-domain alignment constraint between the domain-shared prompt at current stages and the previous incremental stage $P_{pre}$:

$$\mathcal{L}_{IDA} = 1 - \frac{1}{L \times d} \sum_{i=1}^{N} \frac{p_{shared}^i \cdot p_{pre}^i}{\|p_{shared}^i\| \cdot \|p_{pre}^i\|}, \quad (7)$$

$\mathcal{L}_{IDA}$ enhances the ability to learn unified cross-domain feature representations of domain-shared prompt by matching the distribution of them across domains, reducing knowledge loss caused by domain distribution shiftS, and enabling the model to better leverage knowledge from old domains for learning in new domains.

Meanwhile, we introduce an intra-domain orthogonal constraint between the domain-shared prompt and domain-specific prompt to guide them to focus respectively on the learning of general feature

representation and individual domain distribution:

$$\mathcal{L}_{IDO} = \frac{1}{L \times d} \sum_{i=1}^{N} \frac{p_{shared}^i \cdot p_{specific}^i}{\|p_{shared}^i\| \cdot \|p_{specific}^i\|}. \quad (8)$$

Supervision under $\mathcal{L}_{IDO}$ constraint, domain-specific prompt can focus on learning the unique domain distribution information of each domain to facilitate the model in adapting to new domains and preserve this domain characteristic information incorporated into the prompt bank for subsequent inference.

Finally, we can predict the relative depth map $\mathbf{d}_{rel}$ by the frozen pre-trained model $f$ with the learnable complementary domain prompt $\mathbf{P}_{comp}$:

$$\mathbf{d}_{rel} = f(I, \mathbf{P}_{comp}) \quad (9)$$

### 3.4 $S^2$-Adapter

Furthermore, to convert the relative depth map into metric depth map, it is necessary to capture the depth space variations across domains. Here, we decouple the depth space variations into two factors: depth scale and depth shift. The depth scale factor controls the scaling relationship from the relative depth map to the metric depth map, while the depth shift factor indicates the shift between each pixel to the accurate depth value. These two factors are widely used in multi-domain learning to obtain a unified depth representation by metric depth $\hat{\mathbf{d}}$ and stabilize the multi-domain learning process:

$$\mathbf{d}_{rel} = \frac{\hat{\mathbf{d}} - t(\hat{\mathbf{d}})}{s(\hat{\mathbf{d}})}, \quad (10)$$

where $t(\hat{\mathbf{d}}) = \text{median}(\hat{\mathbf{d}})$, and $s(\hat{\mathbf{d}}) = \sum |\hat{\mathbf{d}} - t(\hat{\mathbf{d}})|$. So we can obtain the metric depth map by reversing this process:

$$\hat{\mathbf{d}} = s(\hat{\mathbf{d}})\mathbf{d}_{rel} + t(\hat{\mathbf{d}}). \quad (11)$$

However, as $s(\hat{\mathbf{d}})$ and $t(\hat{\mathbf{d}})$ are unavailable, a direct thought is to estimate them by learning a global scale factor $w$ and shift factor $b$:

$$\hat{\mathbf{d}} = w\mathbf{d}_{rel} + b. \quad (12)$$

Furthermore, due to the potential inaccuracies in the relative depth maps predicted by pre-trained models, a global scale factor and shift factor may not be enough to capture the depth space variations. Therefore, we design a lightweight $S^2$-Adapter which consists of a scale adapter and a shift adapter to learn pixel-wise scale matrix and shift matrix respectively. In practice, we implement each adapter by two convolutional layers with $1 \times 1$ kernel size, followed by the formulation to obtain the final metric depth map:

$$\mathbf{W}_{scale} = ReLU(\mathbf{W}_{1,2}(ReLU(\mathbf{W}_{1,1}(\mathbf{d}_{rel})))), \quad (13)$$

$$\mathbf{W}_{shift} = \mathbf{W}_{2,2}(ReLU(\mathbf{W}_{2,1}(\mathbf{d}_{rel}))), \quad (14)$$

$$\hat{\mathbf{d}} = \mathbf{W}_{scale}\mathbf{d}_{rel} + \mathbf{W}_{shift}, \quad (15)$$

where $W_{i,j}$ means the $j$-th weight of the $i$-th adapter. We employ $ReLU$ activation function to ensure that $\mathbf{W}_{scale}$ is positive, while $\mathbf{W}_{shift}$ is not subject to this constraint. Finally, we can predict the metric depth map through the $S^2$-Adapter:

$$\hat{\mathbf{d}} = S^2\text{-Adapter}(\mathbf{d}_{rel}) \quad (16)$$

And due to the various depth scales in different domains, we use the Scale-Invariant Loss [1] to learn depth knowledge:

$$\mathcal{L}_{\text{Depth}} = \alpha \sqrt{ \frac{1}{T} \sum_i g_i^2 - \frac{\lambda}{T^2} \left( \sum_i g_i \right)^2 }, \tag{17}$$

where $g_i = \log \mathbf{d}_i - \log \hat{\mathbf{d}}_i$ and $\mathbf{d}_i$ denotes the ground truth depth map, and $T$ denotes the number of pixels with valid depth values. We empirically set the $\alpha$ and $\lambda$ to 10 and 0.15 respectively.

Finally, the overall loss $\mathcal{L}$ on each domain is:

$$\mathcal{L} = \mathcal{L}_{\text{Depth}} + \beta \mathcal{L}_{\text{IDO}} + \gamma \mathcal{L}_{\text{IDA}}, \tag{18}$$

where the hyper-parameter $\beta$ and $\gamma$ is empirically set to 1 and 10 respectively.

### 3.5 Domain Prompt Generation

In the inference phase, we devise complementary domain prompt generation to analyze the content of input images to select the corresponding domain-specific prompt and $S^2$-adapter.

In detail, we first extract image features $f_{image} \in \mathbb{R}^{h \times w \times d}$ via a query function that is implemented by the frozen pre-trained model. Then we flatten $f_{image}$ and all domain-specific prompt respectively to get $f'_{image} \in \mathbb{R}^{h*w*d}$ and $f'^{i}_{prompt} \in \mathbb{R}^{N*l*d}$ to compute the inner product between these features and get the domain identity $k$ by:

$$k \leftarrow \operatorname*{argmax}_i (f'_{image} \cdot \left( f'^{i}_{prompt} \right)^T). \tag{19}$$

Consequently, based on this domain identity $k$, we can generate the complementary domain prompt and inject it into the model. Additionally, we can select the $S^2$-Adapter based on the domain identity $k$. Thus, the corresponding complementary domain prompt and the $S^2$-Adapter are accurately selected to guide the model for depth predictions.

## 4 EXPERIMENTS

### 4.1 Experimental Setup

**Datasets.** We evaluate our method on 12 benchmark datasets, including 2 indoor and 10 outdoor datasets with different characteristics. The details of them are given in Table 1, and the data augmentation strategies are described in the supplementary material.

**Implementation details.** We implement experiments on PyTorch and train with RTX 3090 GPU with the AdamW optimizer for 10 epochs in each domain. We use a batch size of 8 and a one-cycle policy schedule for adjusting the learning rate with a max learning rate of 0.0000161. We use the dpt-hybrid-based model from work [29] as our backbone. We evaluate our method with 5 metrics: relative mean absolute error (AbsRel); Root Mean Squared Error (RMSE); threshold accuracy $\delta_{1.25}$; Average metrics and Forgetting metrics. The Average Metric represents the generalization capability of the model and is calculated by the mean value of the RMSE metric across all domains in the final incremental session. The Forgetting Metric assesses the ability of the model against forgetting and represents the difference between the RMSE metric of the first domain after all training stages and its value after the completion of the first stage of training.

Table 1: Details of datasets involved in the experiments.

| Datasets | Characteristic | Range (m) | #Samples (Train/Test) |
|---|---|---|---|
| NYU_v2 [35] | Indoor | 0~10 | 50K/0.6k |
| ScanNet [8] | Indoor | 0~6 | 50k/17k |
| KITTI [38] | Outdoor | 0~80 | 85k/1k |
| vKITTI_v2 [4] | Synthetic | 0~80 | 12k/0.6k |
| Cityscapes [7] | Outdoor | 0~200 | 2.9k/0.5k |
| CS_Foggy [32] | Foggy | 0~200 | 8.9k/1.5k |
| CS_Rainy [13] | Rainy | 0~200 | 9.5k/1.1k |
| DAOD [45] | Outdoor | 0~120 | 174k/7.7k |
| D_Sunny [45] | Sunny | 0~120 | 0.4k/0.1k |
| D_Cloudy [45] | Cloudy | 0~120 | 0.4k/0.1k |
| D_Foggy [45] | Foggy | 0~120 | 0.4k/0.1k |
| D_Rainy [45] | Rainy | 0~120 | 0.4k/0.1k |

**Incremental domain sequences.** We design 4 incremental domain sequences to comprehensively validate the performance of the model:

(1) **NYU_v2→ScanNet→KITTI.** In this sequence, the depth range varies cross-domain (10m→6m→80m). Therefore, this sequence is primarily employed to assess the adapting capacity of models to depth space variations.

(2) **vKITTI_v2→KITTI→CityScapes.** The primary focus of this sequence is to assess the model's capacity to adapt to image style variations, such as from virtual to real scenes.

(3) **CityScapes→CS_Foggy→CS_Rainy.** In this sequence, the weather conditions in each domain are different. Therefore, training under this sequence can verify the capability of models to mitigate domain distribution shifts.

(4) **DAOD→D_Sunny→D_Cloudy→D_Foggy→D_Rainy.** Due to the extreme imbalance of the sample numbers across domains in this sequence, this sequence can test the model's adaptability and resistance to forgetting in long-term few-shot scenarios.

**Comparisons methods.** We compare DSSP against our baseline, the classical incremental learning method (EWC [17]), and the SOTA incremental monocular depth estimation method (LL-MonoDepth [11]) under the above four incremental domain sequences. To evaluate the performance of our baseline, we train it through continuous fine-tuning (FT) and joint-domain training (JDT). We implement the classical incremental learning method EWC [17] based on our backbone. To reproduce the results of the LL-MonoDepth, we retain 500 replay samples for it in each domain, consistent with the original paper. Notably, due to the insufficient number of training samples under other weather conditions in DAOD dataset, we randomly retain 10% of the training samples for replaying in these domains.

### 4.2 Quantitative Results

**Comparisons under different depth ranges and image styles.** As illustrated in Table 2, DSSP achieves superior performance under all incremental domain sequences. Specifically, in the first sequence, compared to the SOTA methods, DSSP exhibits a significant improvement in the Average metric, demonstrating that our model

**Table 2: The quantitative comparison with SOTA methods on three incremental settings which include "NYU_v2→ScanNet→KITTI", "vKITTI_v2→KITTI→CityScapes" and "CityScapes→CityScapes_Foggy→CityScapes_Rainy".**

| Methods | NYU_v2 | | | ScanNet | | | KITTI | | | Average ↓ | Forgetting ↓ |
|---|---|---|---|---|---|---|---|---|---|---|---|
| | AbsRel ↓ | RMSE ↓ | $\delta_{1.25}$ ↑ | AbsRel ↓ | RMSE ↓ | $\delta_{1.25}$ ↑ | AbsRel ↓ | RMSE ↓ | $\delta_{1.25}$ ↑ | | |
| FT | 2.210 | 5.161 | 0.656 | 2.494 | 3.748 | 0.426 | 0.194 | 4.361 | 74.707 | 4.423 | 4.759 |
| JDT [29] | 2.091 | 5.260 | 0.028 | 1.702 | 3.169 | 0.079 | 0.260 | 5.048 | 4.492 | 15.018 | - |
| EWC [17] | 2.304 | 5.367 | 0.674 | 2.497 | 3.776 | 0.424 | 0.191 | 4.323 | 77.758 | 4.489 | 4.960 |
| LL-MonoDepth [11] | 0.224 | 0.810 | 56.675 | 0.259 | 0.526 | 52.580 | 0.193 | 11.365 | 66.792 | 4.234 | 0.277 |
| **DSSP (Ours)** | 0.276 | 0.716 | 56.528 | 0.120 | 0.247 | 86.083 | 0.216 | 4.795 | 66.469 | **1.919** | **0.255** |

| Methods | vKITTI_v2 | | | KITTI | | | CityScapes | | | Average ↓ | Forgetting ↓ |
|---|---|---|---|---|---|---|---|---|---|---|---|
| | AbsRel ↓ | RMSE ↓ | $\delta_{1.25}$ ↑ | AbsRel ↓ | RMSE ↓ | $\delta_{1.25}$ ↑ | AbsRel ↓ | RMSE ↓ | $\delta_{1.25}$ ↑ | | |
| FT | 6.474 | 42.982 | 0.161 | 1.968 | 33.161 | 5.818 | 17.008 | 16.927 | 54.934 | 31.023 | 39.900 |
| JDT [29] | 1.449 | 14.064 | 1.447 | 0.190 | 4.331 | 77.576 | 21.448 | 13.711 | 90.434 | 10.702 | - |
| EWC [17] | 4.791 | 37.422 | 0.604 | 1.613 | 28.243 | 9.512 | 17.020 | 15.751 | 59.803 | 27.139 | 34.251 |
| LL-MonoDepth [11] | 0.354 | 9.376 | 44.932 | 0.407 | 15.059 | 23.887 | 10.753 | 45.718 | 20.320 | 23.384 | 5.171 |
| **DSSP (Ours)** | 1.119 | 7.909 | 13.590 | 0.131 | 4.517 | 86.379 | 22.503 | 17.464 | 52.625 | **9.963** | **4.279** |

| Methods | CityScapes | | | CityScapes_Foggy | | | CityScapes_Rainy | | | Average ↓ | Forgetting ↓ |
|---|---|---|---|---|---|---|---|---|---|---|---|
| | AbsRel ↓ | RMSE ↓ | $\delta_{1.25}$ ↑ | AbsRel ↓ | RMSE ↓ | $\delta_{1.25}$ ↑ | AbsRel ↓ | RMSE ↓ | $\delta_{1.25}$ ↑ | | |
| FT | 24.300 | 14.213 | 89.352 | 24.959 | 14.224 | 89.455 | 3.726 | 10.259 | 92.062 | 12.898 | -1.506 |
| JDT [29] | 19.539 | 16.731 | 56.898 | 26.241 | 16.445 | 57.310 | 2.847 | 15.218 | 51.943 | 16.131 | - |
| EWC [17] | 14.964 | 16.226 | 58.889 | 14.626 | 15.924 | 59.147 | 3.032 | 14.798 | 53.804 | **15.649** | **-1.450** |
| LL-MonoDepth [11] | 10.440 | 16.819 | 68.531 | 10.781 | 15.957 | 69.962 | 5.379 | 24.115 | 56.459 | 18.964 | -0.389 |
| **DSSP (Ours)** | 27.851 | 17.274 | 59.645 | 12.775 | 15.210 | 61.360 | 3.789 | 16.364 | 48.647 | 16.282 | -0.295 |

effectively mitigates the influence of the depth space variations through $S^2$-Adapters. In the second sequence, benefiting from our complementary domain prompt, DSSP is capable of learning unified feature representations across domains while also acquiring individual domain distributions for each domain. Therefore DSSP can maintain superior performance in both Average and Forgetting metrics. In the third sequence, none of the methods exhibited any occurrence of forgetting (i.e., forgetting metrics are less than 0). This indicates that under this scenario, the domain shift induced by weather variations is not as much as in the former two sequences, thereby enabling satisfactory performance through direct fine-tuning (FT).

Figure 4 illustrates a comparison between our DSSP and other methods across all domains and shows the final results. It can be

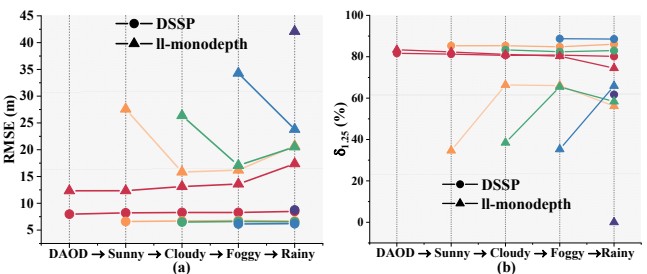

**Figure 3: Comparative analysis of DSSP and the SOTA incremental monocular estimation methods (LL-MonoDepth) on mitigating forgetting performance under the few-shot DAOD incremental setting.**

observed that nearly all methods produce satisfactory depth map predictions in the last domain (KITTI). Notably, benefiting from complementary domain prompt, which alleviates the effects of domain distribution shifts, DSSP maintains extraordinary performance in all domains. Meanwhile, owing to the inclination of this incremental sequence to validate the model's adaptability across varying depth ranges, our model, facilitated by $S^2$-Adapter, exhibits the capacity to adapt depth space variations. For instance, within the first two indoor datasets, our model exhibits a relatively accurate measure of the depth of distant objects. The qualitative results under the other three additional incremental settings are provided in the supplementary material.

**Comparisons under the long-term few-shot scenario.** Here, we conduct experiments on long-term few-shot scenarios, where the model is required to adapt 5 domains with different weathers. And the training samples of new domains are only 400. Figure 3 describes the performance curve of different methods during training. As one can observe, DSSP is more stable and achieves better performance in each session. As illustrated in Table 3, DSSP achieves outstanding performance compared with other methods. Compared with EWC, DSSP obtains an improvement of 2.569 in terms of average RMSE over all domains. We also notice that DSSP is slightly inferior to EWC in terms of forgetting rate. Because EWC unfrozes the large-scale pre-trained model with additional regularization. Such a paradigm consumes plenty of time and computational resources and may destroy the pre-trained knowledge. Compared with the SOTA incremental depth estimation method LL-MonoDepth, DSSP obtains significant improvements in terms of Average and Forgetting metrics. The superior performance can be attributed to two

**Table 3: The quantitative results on "DAOD→D_Sunny→D_Cloudy→D_Foggy→D_Rainy" incremental settings.**

| Methods | DAOD | | | DAOD_Sunny | | | DAOD_Cloudy | | | DAOD_Foggy | | | DAOD_Rainy | | | Average ↓ | Forgetting ↓ |
|---|---|---|---|---|---|---|---|---|---|---|---|---|---|---|---|---|---|
| | AbsRel ↓ | RMSE ↓ | $\delta_{1.25}$ ↑ | AbsRel ↓ | RMSE ↓ | $\delta_{1.25}$ ↑ | AbsRel ↓ | RMSE ↓ | $\delta_{1.25}$ ↑ | AbsRel ↓ | RMSE ↓ | $\delta_{1.25}$ ↑ | AbsRel ↓ | RMSE ↓ | $\delta_{1.25}$ ↑ | | |
| FT | 0.320 | 9.045 | 27.116 | 0.365 | 9.506 | 13.992 | 0.392 | 9.980 | 8.563 | 0.345 | 10.331 | 14.391 | 0.306 | 10.322 | 28.224 | 9.837 | 0.080 |
| JDT [29] | 0.330 | 8.944 | 23.696 | 0.367 | 9.273 | 13.310 | 0.395 | 9.609 | 8.820 | 0.355 | 10.653 | 13.296 | 0.307 | 10.097 | 29.566 | 9.715 | - |
| EWC [17] | 0.313 | 9.013 | 28.951 | 0.359 | 9.391 | 15.315 | 0.381 | 9.898 | 9.739 | 0.340 | 10.311 | 15.209 | 0.302 | 10.733 | 30.288 | 9.869 | **-0.212** |
| LL-MonoDepth [11] | 0.170 | 17.379 | 74.500 | 0.333 | 20.694 | 56.131 | 0.325 | 20.523 | 58.340 | 0.263 | 23.806 | 65.896 | 0.999 | 42.089 | 0.000 | 24.898 | 5.045 |
| **DSSP (ours)** | 0.177 | 8.494 | 80.171 | 0.130 | 6.681 | 86.041 | 0.165 | 6.534 | 82.974 | 0.148 | 6.186 | 88.570 | 0.226 | 8.754 | 61.708 | **7.330** | 0.524 |

folds. Firstly, our domain-shared prompt can learn the universality across domains in feature spaces, reducing the domain gap during incremental learning. Secondly, $S^2$-Adapter and domain-specific prompt can learn and store the domain distribution, improving the adaptability of the model for each domain.

## 4.3 Ablation Studies

**Shared and Specific Prompt**. We first examine our DSSP by replacing domain-shared prompt and domain-specific prompt respectively. As illustrated in Table 4 (*w/o* $P_{shared}$ *and w/o* $P_{specific}$). The model suffers from serious forgetting problems when the model can only learn unified feature representations between domains and lacks domain-specific knowledge which demonstrates the enhancing effect of domain-specific prompt on the adaptability of the model.

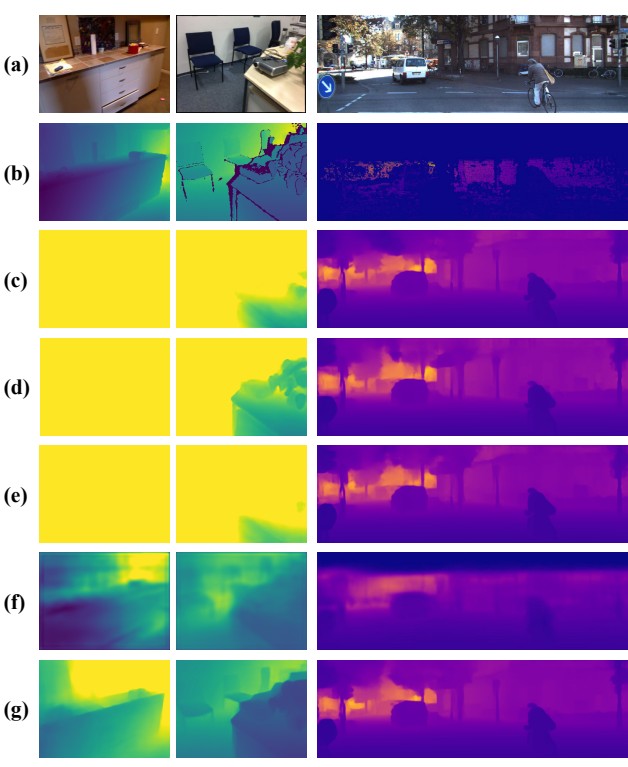

**Figure 4: Qualitative comparison with SOTA methods in the learning order of NYU-v2 → ScanNet → KITTI. (a) RGB. (b) Ground truths. (c) FT. (d) JDT. (e) EWC. (f) LL-MonoDepth. (g) Ours.**

$S^2$-**Adapter**. Then, we access the performance of our DSSP without the $S^2$-Adapter. As shown in Table 4 (*w/o* $S^2$-*Adapter*), the absence of the $S^2$-Adapter leads to the lack of capability of adapting the depth space variations thereby producing the severe forgetting problem.

**Regularization Loss**. We further validate the effects of the constraints imposed on domain-shared prompt and domain-specific prompt. As demonstrated in Table 4 (*w/o* $\mathcal{L}_{IDO}$, *w/o* $\mathcal{L}_{IDA}$ *and w/o all* $\mathcal{R}$ ), the absence of either constraint leads to a decrease in the model's generalization ability and resilience to forgetting. Specifically, the lack of inter-domain alignment constraint for domain-shared prompt hinders the learning of unified feature representations across domains, thereby failing to ease domain gaps, ultimately resulting in severe forgetting phenomena. Similarly, the absence of intra-domain orthogonality constraint for both types of prompts induces knowledge interference during learning, preventing domain-specific prompt from acquiring individual domain distribution information, thereby impeding the model's adaptation to new domains and weakening its generalization capability.

**The length of the Prompt**. In Figure 5, we validate the effect of prompt length on "NYU-v2 → ScanNet → KITTI" task. As one can observe, the best Average is achieved when the prompt length is 300. The best Forgetting is achieved when the prompt length is 250. For considerations of storage and efficiency, we adopt a default prompt length of 100.

## 4.4 Domain Identity Prediction Results.

Then we study the prediction accuracy of domain identity by our method. As shown in Figure 6, our method achieves good performance under the "NYU_v2, ScanNet, KITTI" sequences which demonstrates that our domain-specific prompt can learn the individual domain distribution information. Meanwhile, due to the small number of samples in the DAOD incremental sequences, our method can accurately predict the majority of domains to which the samples belong.

**Table 4: The ablation studies of each components in DSSP on NYU_v2->ScanNet->KITTI incremental setting.**

| Methods | NYU_v2 | | | Average ↓ | Forgetting ↓ |
|---|---|---|---|---|---|
| | AbsRel ↓ | RMSE ↓ | $\delta_{1.25}$ ↑ | | |
| w/o $P_{shared}$ | 0.175 | 0.577 | 73.221 | 1.670 | - |
| w/o $P_{specific}$ | 1.658 | 3.750 | 8.153 | 2.776 | 3.173 |
| w/o $S^2$-Adapter | 3.360 | 6.978 | 0.628 | 3.884 | 5.178 |
| w/o $\mathcal{L}_{IDO}$ | 0.481 | 1.028 | 31.066 | 5.042 | 0.586 |
| w/o $\mathcal{L}_{IDA}$ | 1.558 | 3.316 | 2.768 | 5.663 | 2.869 |
| w/o all $\mathcal{R}$ | 1.187 | 2.477 | 5.504 | 5.823 | 2.036 |
| Ours | 0.276 | 0.716 | 56.528 | **1.919** | **0.255** |

                                                                  Anonymous Authors

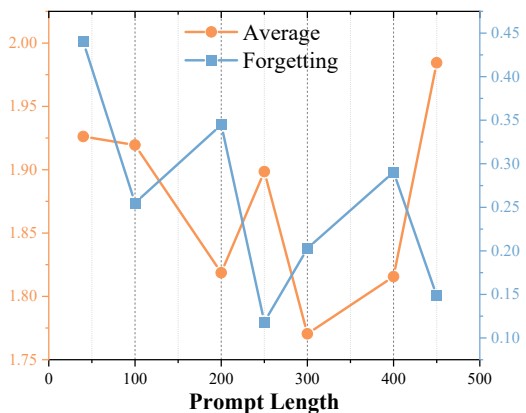

Figure 5: Effect of the length of our complementary domain prompt measured by the Average and Forgetting metrics in the learning order of NYU-v2 → ScanNet → KITTI.

## 4.5 Ablation of the Incremental Domain Orders

Due to varying degrees of inter-domain gaps, different incremental domain orders may result in diverse results. Consequently, we evaluate the performance of our method across all sequences on three datasets: NYU_v2, ScanNet, and KITTI. The performance of our method under different sequences on other datasets and detailed results of metrics are presented in the supplementary material. As shown in Figure 7, due to the substantial disparity in depth range between the KITTI dataset and other indoor datasets, training a model solely on KITTI and subsequently on other datasets leads to significant performance degradation (a)(f). However, when the

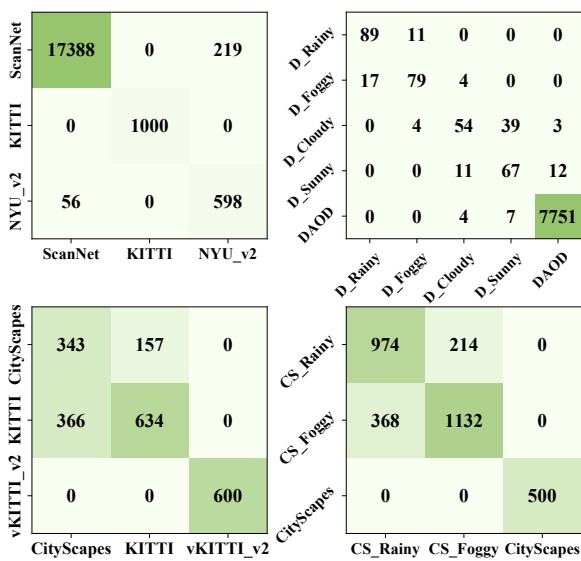

Figure 6: The accuracy of the domain identity predictions on four incremental sequences. The horizontal axis represents our prediction results, and the vertical axis represents the true domain identity.

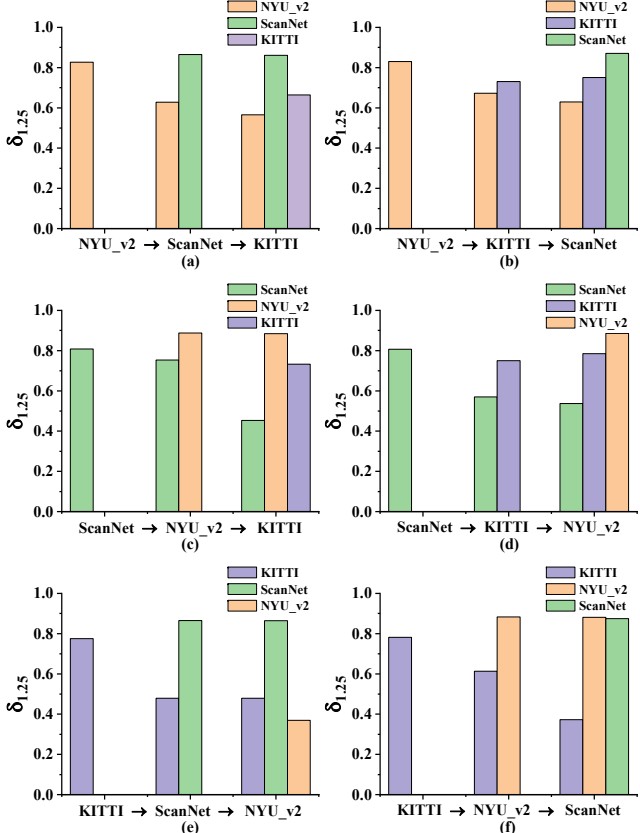

Figure 7: The $\delta_{1.25}$ accuracy of DSSP under different incremental domain orders of NYU_v2, ScanNet and KITTI datasets.

model has been pre-trained on any indoor dataset, its performance on the KITTI dataset remains stable or even improves (b)(d).

## 5 CONCLUSION

In this paper, we develop the DSSP for incremental monocular estimation without data privacy and security concerns, which designs complementary domain prompt and $S^2$-Adapter to mitigate domain distribution shifts and depth space variations. Specifically, we first generate the complementary domain prompt by learning domain shared and specific prompt which are supervised by the inter-domain alignment and intra-domain orthogonal constraint to facilitate the model to predict a unified depth representation. Then, we design a lightweight $S^2$-Adapter that quantizes the depth variations into scaling and shifting, enhancing the ability of the model to capture the mapping from unified depth representations to metric depth space and eventually predict the metric depth map. Extensive experiments on 12 public datasets under four different incremental scenarios demonstrate the superiority of DSSP.

We design four incremental domain sequences to provide a comprehensive benchmark for incremental monocular depth estimation. In the future, we will validate the forgetting issue and the transfer ability of our DSSP on larger depth estimation pre-trained models.

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
