# OpenReview forum: "Domain Shared and Specific Prompt Learning for Incremental Monocular Depth Estimation"
_acmmm.org/ACMMM/2024/Conference — MM2024 Poster_

### Official Review · Reviewer_9MXC · 2024-05-01

**Rating:** 3
**Confidence:** 3

**Summary:**

This paper proposes to solve the incremental monocular depth estimation problem via prompt learning  without replaying samples. The proposed Domain Shared and Specific Prompt Learning (DSSP) adopts complementary domain prompts to mitigate domain distribution shifts and depth space variations. The prompt learning helps to learn task-specific knowledge in the new domain. The proposed method is verified under various scenarios under different conditions.

**Strengths:**

1. The writing of the paper is clear and easy to follow, with few issues (detailed in the weaknesses section).

2. This paper is theoretically well-motivated and the results are impressive compared to other attempts.

**Limitations:**

1. What is the prompt bank size and how the prompt bank is updated and will the memory mechanism brings in large computation cost?

2. The data of different domains are incrementally obtained. Specifically, when obtaining data from different domains, is it considered to assign different weights? Will such a design affect the performance in certain domains?

3. What is the purpose to use metric depth and will it brings more befenit than relative depth?

4. How much improvement can Shared and Specific Prompts each bring, and is there any related ablation study?

5. The method in this paper significantly improves performance on some datasets, such as Cityscapes, but shows weaker improvement on datasets like NYU and KITTI. How can this phenomenon be explained, and does it suggest that the method might overfit?

**Suitability:**

2

---

### Official Review · Reviewer_3JwA · 2024-05-23

**Rating:** 3
**Confidence:** 2

**Summary:**

The paper proposes a method called Domain Shared and Specific Prompt Learning (DSSP) for incremental monocular depth estimation, aiming to continuously learn from new domains while maintaining performance on old domains. The method addresses the catastrophic forgetting problem caused by domain distribution shifts and depth space variations. DSSP uses complementary domain prompts and a Scale&Shift Adapter (S2-Adapter) to mitigate these issues without requiring replay samples, thus preserving data privacy and security. The paper demonstrates that DSSP achieves state-of-the-art performance across various scenarios, including different depth ranges, virtual and real environments, weather conditions, and few-shot incremental learning on 12 datasets.

**Strengths:**

- The paper includes extensive experiments across multiple datasets and scenarios, demonstrating the effectiveness of DSSP.

- The paper is well-organized and clearly written, with detailed descriptions of the methodology, experimental setup, and results. Figures and tables are used effectively to illustrate the concepts and findings.

**Limitations:**

- The paper mentioned that the DSSP can reduce storage costs. However, this assertion requires further validation with results.
- You should present the results of progressively adding each component of the method and displaying the associated results at each step until the final method is achieved. This differs from the ablation study presented in Table 4.
- Why can the pre-trained model with M transformer layers obtain M domain-shared / -specific prompts? You should provide more details.
- The proposed method involves multiple components, including domain-shared and specific prompts, S2-Adapter, and various loss functions, which might increase the complexity of both training and testing. Further discussion regarding training and inference times would be beneficial.

**Suitability:**

2

---

### Official Review · Reviewer_ybV6 · 2024-05-25

**Rating:** 4
**Confidence:** 3

**Summary:**

The main content of this paper is the research on incremental monocular depth estimation, which aims to solve the catastrophic forgetting problem that models face when adapting to new domains. The paper puts forward two main contributions:
Domain Shared and Specific Prompt Learning (DSSP) : A new incremental monadic depth estimation method is proposed, which mitigated the problems of domain distribution shifts and depth space variations by designing complementary domain prompts. This approach solves data privacy and security concerns by eliminating the need to store original samples of old domains
S2-Adapter: An adapter is designed to quantify the variation of depth space, which converts domain shared depth space into domain specific depth space by scaling and shifting matrix. This adapter enhances the ability of the model to capture mapping from a unified depth representation to a metric depth space, thereby improving the predictive accuracy of the final metric depth map.
And the paper has completed enough experiments to prove its effectiveness.

**Strengths:**

The experimental Settings of the paper are sufficient. Extensive experiments were carried out on 12 datasets covering different scenarios, such as different depth ranges, virtual and real environments, and different weather conditions, which demonstrated the robustness and applicability of the method. Four incremental domain sequences are designed to comprehensively verify the performance of the model. This comprehensive evaluation strategy helps to demonstrate the effectiveness of the method in different situations.
In addition, the technical details described in the article are sufficient, and detailed mathematical formulas and algorithm descriptions help readers understand the working principle of the method and can reproduce the experimental results.

**Limitations:**

1, the paper uses Prompt Bank to implement the existing domain review. However, in the acquisition of existing knowledge, common methods include Latent Representation-based RAG, etc., which can further explain the motivation of using this structure.
2. In Section 3 MYTHOLOGY, the specific details about the network are not explained, including the size of input and output, the number of prompt, dimension and so on.

**Suitability:**

2

---

### Meta-Review · Area_Chair_mT6d · 2024-07-02

**Recommendation:** Accept (Poster)
**Confidence:** 4

**Metareview:**

This paper presented a method for incremental monocular depth estimation, aiming to address the problem of catastrophic forgetting when adapting to new domains. Specifically, a domain shared and specific prompt learning (DSSP) approach was proposed to achieve the goal. The paper received comments from three reviewers. The main strengths of this work are: the extensive experimental analysis, and the writing. On the other hand, there are some concerns raised by the reviewers, including: insufficient details about the methodology and some statements, the motivation of particular structure designs, missing explanation of some weak performance, and concerns about the domain-shared and domain-specific design justification.

Overall, most concerns were addressed during the rebuttal, though there are still some remain. Considering the remaining concerns are relatively minor and can be addressed with a minor revision, the AC would like to recommend Accept, but strongly suggest the authors include the supporting evidence for those raised concerns in the final version by carefully reviewing again the comments from all the reviewers.